


# An Investigation into the Relationship between Teleconnections and Taiwan's Streamflow

Chia-Jeng Chen[1] and Tsung-Yu Lee[2]

[1]National Chung Hsing University, 145 Xingda Road, Taichung, Taichung 40227, Taiwan
[2]National Taiwan Normal University, 162 Heping East Road, Section 1, Taipei 10610, Taiwan

*Correspondence to:* Tsung-Yu Lee (tylee@ntnu.edu.tw)

**Abstract.** Interannual variations in catchment streamflow represent an integrated response to anomalies in regional moisture transport and atmospheric circulations and are ultimately linked to large-scale climate oscillations. This study investigates the relationship between Taiwan's long-term summertime (July to September, JAS) streamflow and manifold teleconnection patterns. Lagged correlation analysis is conducted to calculate how JAS streamflow data derived at 28 upstream and 13 down-
stream gauges in Taiwan correlate with 14 teleconnection indices in the current or preceding seasons. Of the many indices, the West-Pacific and Pacific-Japan (PJ) patterns, both of which play a critical role in determining cyclonic activity in the western North Pacific basin, exhibit the highest concurrent correlations (most significant $r = 0.50$) with the JAS flows in Taiwan. Alternatively, the Quasi-Biennial Oscillation averaged over the period from the previous October to June of the current year is significantly correlated with the JAS flows (most significant $r = -0.66$), indicating some forecasting utility. By further exam-
ining the correlation results using a 20-year moving window, peculiar temporal variations and possible climate regime shifts (CRSs) can be revealed. To identify suspicious, abrupt changes in the correlation, a CRS test is employed. The late 1970s and 1990s are identified as two significant change points, and during the intermediate period, a marked in-phase relationship ($r > 0.8$) between Taiwan's streamflow and the PJ index is observed. Linear regression models that incorporate the climate indices into streamflow prediction are found to provide reasonable prediction skill in general, and the models are then used to
illustrate the dramatic variations in prediction skill from the pre- to post-regime shift epoch. It is verified that the two shifts are in concordance with the alteration of large-scale circulations in the Pacific basin. The changes in pattern correlation and composite maps before and after the change point are discussed, and our results suggest that empirical forecasting techniques should take into account the effect of CRSs on predictor screening.

## 1 Introduction

Hydro-climatic forecasting is a crucial issue, particularly for those regions suffering increased rainfall intensity and/or reoccurring, persistent droughts in a changing climate. An established method for hydro-climatic forecasting is the usage of "teleconnection" patterns (van den Dool, 2007) that signify the influence of low-frequency climate oscillations on hydro-climates in remote locations by emanating shifts in meteorological systems (e.g., planetary waves, jet streams, and monsoons). Prominent teleconnection patterns have proven useful for the prediction of regional climates with lead times from weeks to months (e.g.,





Palmer and Anderson, 1994; Chiew et al., 1998; Goddard et al., 2001). In the equatorial Pacific, El Niño-Southern Oscillation (ENSO) stands out as the leading mode and has dramatic impacts on global and regional climates (Ropelewski and Halpert, 1986, 1987, 1989, 1996; Kiladis and Diaz, 1989; Harrison and Larkin, 1998; Dai and Wigley, 2000; McCabe and Dettinger, 1999; Wang et al., 2000). In recent years, discussion regarding non-canonical ENSO (e.g., Central-Pacific El Niño or El Niño

Modoki, Ashok et al., 2007) and associated impacts has stimulated a significant increase in ENSO-related research. Because ENSO has been studied for decades, for quantitative and monitoring purposes, researchers have defined several ENSO indices, including NINO 1+2, NINO 3, NINO 4, and NINO 3.4 (e.g., Trenberth, 1997; Trenberth and Stepaniak, 2001).

In addition to ENSO, many other teleconnection patterns over fields of atmospheric variables (e.g., sea level pressure and 500-mb geopotential height) have been recognized by scientists using various statistical techniques, such as correlation analysis

by Wallace and Gutzler (1981) and rotated principal component analysis (RPCA) by Barnston and Livezey (1987). Prominent teleconnection patterns include the North Atlantic Oscillation (NAO, Hurrell, 1995), Pacific North American (PNA, Wallace and Gutzler, 1981; Barnston and Livezey, 1987), Indian Ocean Dipole (IOD, Saji et al., 1999), West Pacific (WP, Barnston and Livezey, 1987), East Pacific-North Pacific (EP-NP, Barnston and Livezey, 1987), Pacific-Japan (PJ, Nita, 1987; Kosaka and Nakamura, 2006), Artic Oscillation (AO, Thompson and Wallace, 1998), and Antarctic Oscillation (AAO, Gong and Wang,

1999). If a high-frequency filter is applied to atmospheric/oceanic variables of interest (i.e., to reserve the low-frequency component) prior to pattern recognition, other patterns referred to as "interdecadal modes" can be observed. For example, the Pacific Decadal Oscillation (PDO, Mantua et al., 1997) and Atlantic Multidecadal Oscillation (AMO, Schlesinger and Ramankutty, 1994; Enfield et al., 2001) both shift phases with a period longer than several decades. All of the above interannual and interdecadal oscillations have been shown to have widespread impacts on regional and global climate systems.

Several studies (e.g., Wang et al., 2000; Yang et al., 2002; Wang and Fan, 2005; Choi et al., 2012) have investigated the various effects of teleconnection patterns on East Asian regions, in which the island country of Taiwan, with an area of approximately 36,000 km$^2$, is situated (Figure 1). In Taiwan, prevailing weather systems found in East Asia can also be observed (Yihui and Chan, 2005; Chen and Chen, 2011), including spring rains, Mei-Yu, and East Asian monsoons from spring to summer, typhoons from summer to autumn, and the Mongolian high pressure system and associated northeast monsoons in winter.

Because of the Central Mountain Range (topographic variations) and gradually varied climate zones (latitudinal differences), the influence of those weather systems on precipitation in particular can show great east-west and north-south contrasts. As a result, while the wet season generally spans from summer to autumn based on the long-term average, Taiwan's precipitation and streamflow in the wet season exhibits great spatial distributions of prominent intra-seasonal and inter-annual variations. Thus, the search for the relationship between Taiwan's climate in the wet season and large-scale circulations can guide the

development of a hydro-climatic forecasting framework potentially of benefit to water resource management in this area.

Using teleconnection patterns to predict regional precipitation is straightforward, as precipitation is a hydro-climatic variable signifying the regional sink of large-scale moisture transport. However, precipitation forecasting can be largely hampered by its spatio-temporal heterogeneity, mainly attributed to the influence of rugged landforms (as the characteristic topography in Taiwan; Lee et al., 2015). Because of the interaction between topography and distinct synoptic weather systems among

seasons, the challenge of precipitation forecasting in Taiwan can easily escalate. Alternatively, streamflow data measured at





a watershed outlet represent an *integrated* response to spatial and temporal precipitation distribution within the watershed, largely attenuating the influence of precipitation heterogeneity. In addition, given the assumption that evapotranspiration can be neglected during the wet season (e.g., summer in Taiwan), the streamflow variable can be an ideal surrogate for interpreting long-term climate variability. The explicit linkage between streamflow and water resources and hydro-meteorological hazards

also indicates that effective streamflow prediction can produce immediate utility for a variety of users.

In fact, the usage of teleconnection patterns for streamflow forecasting is enlightened by a number of previous endeavours. Earlier work can be exemplified by Kahya and Dracup (1993) and Hamlet and Lettenmaier (1999); whereas the former examined the relationship between ENSO and the unimpaired streamflow over the contiguous United States, the latter devised an empirical model for forecasting of the Columbia River streamflow using ENSO and PDO. Another earlier work by Chiew et al.

(1998) linked ENSO to the Australian streamflow, and then Chiew and McMahon (2002) later extended their discussion to global ENSO-streamflow teleconnection. More recently, Moradkhani and Meier (2010) adopted various climate indices along with several climate variables and then employed a principal component regression model for streamflow forecasting in two Pacific Northwest basins; Robertson and Wang (2012) applied a Bayesian approach to select predictors from a pool of 13 climate indices for seasonal streamflow forecasting in Australia; Hidalgo-Muñoz et al. (2015) used multiple linear regression

with 20 teleconnection indices as potential predictors to forecast seasonal streamflow over the Iberian Peninsula. From these studies, it is noted that, while the usage of a comprehensive list of climate indices appears to be the trend, most studies have focused on coping with modern statistical techniques and pursuing optimal skill rather than diagnosing underlying mechanisms of predictability and noting caveats of intrinsic covariability between regional streamflow and large-scale circulations. Furthermore, to the best of our knowledge, similar work applied to Taiwan or most East Asian regions has not yet been conducted.

The above rationale therefore serves as the backbone of our research opportunity.

One of the caveats of using teleconnection patterns for hydro-climatic forecasting that should be addressed is the existence of climate regime shifts (CRS). The climate system, as demonstrated by the phase changes in the PDO and AMO, can undergo a reconfiguration, shifting its state from one to another. A steady climate regime can last for decades, but a CRS usually occurs momentarily (e.g., in a particular year or so). The occurrence of a CRS indicates not only a new climate state but also

deterioration or even a possible break-off of the relationship between regional hydro-climates and certain circulation patterns. Many researchers have identified notable CRSs in the Pacific basin (e.g., in the late 1970s by Miller et al., 1994, and Hare and Mantua, 2000, and in the late 1990s by McPhaden et al., 2011, and Hong et al., 2014a). However, neither the shift in the correlation of Taiwan's streamflow with teleconnection patterns nor the impact of the CRS on streamflow forecasting has been thoroughly discussed.

The above introduction explains the main motivation for our work, which comprises three primary objectives:

1. To investigate the relationships between Taiwan's streamflow and teleconnection patterns by conducting correlation analysis between seasonal streamflow data and major climate indices;

2. To verify the existence of any CRS signals in the correlation and discuss associated changes in large-scale circulation patterns; and,





3. To illustrate the overall prediction skill and the effect of CRSs on streamflow prediction in Taiwan using a linear regression approach.

The rest of this paper is organized as follows. Section 2 describes the data and analysis procedures used in this study. To align with the study objectives, Sections 3 presents the results of correlation and CRS analyses and then shows the variations in prediction skill due to the shifts using linear regression. Section 4 further discusses the implication of CRSs for seasonal forecasting. Lastly, Section 5 provides a summary of our findings and concluding remarks.

## 2   Data and Analysis Procedures

As this study aims to examine how Taiwan's streamflow is related to large-scale circulations, its geographical location, major watersheds and sub-catchments are shown in Figure 1. Streamflow data in the watersheds will be correlated with selected climate indices. In the subsections below, the specifications and sources of the streamflow, climate index, and other auxiliary data will be amply described, followed by the analysis procedures used in this study.

### 2.1   Streamflow data, climate indices, and other data sets

Streamflow data used in this study are obtained from the Water Resources Agency in Taiwan, the primary authority in charge of installing and monitoring most of the river gauges over the country, and from the Taiwan Power Company, measuring streamflow for the sake of hydroelectricity. Out of many gauges possessed by these agencies, a total of 28 upstream and 13 downstream gauges of satisfactory quality and extended record are selected. The collective contributing area associated with those downstream gauges located at the outlets of 13 major watersheds is approximately 16,731 km$^2$, ~46% of Taiwan's territory. Because streamflow data observed at the downstream gauges are subject to human intervention (e.g., water regulation and withdrawal for various consumption), a distinction has been made to separate those upstream gauges with pristine flows from the downstream gauges, as shown in Figure 1. From here onward, two batches of the same (or similar) analysis will be performed for the upstream and downstream data. However, to minimize the effect of human intervention on streamflow data (and to still include those downstream data in our analysis), the scope of this study is designed to emphasize data during the high-flow season (as frequent regulations usually occur during low-flow seasons). Without loss of generality, July to September (JAS) is identified as the target high-flow season, from which seasonal streamflow data are aggregated for our follow-up analysis. JAS is also known as the major typhoon season in Taiwan, so our analysis will present implications reflecting typhoon activity to some extent. Here, the simplest strategy is adopted for human intervention avoidance, but some alternatives to reconstructing natural flows (e.g., Wen, 2009) do exist and could facilitate our ongoing analysis during other seasons.

The periods of record and the missing data percentage for all 41 catchments are listed in Table 1. Correlation analysis typically requires a sufficiently long period of record. Thus, we decide to use all available data even though their periods of record are not entirely the same. 30 out of 41 gauges present less than 3% missing data (e.g., 1 out of 40 years is missing), indicating that the quality of JAS flow data is quite reasonable. For those missing-data years, we do not perform any data filling





because we do not want to create any artificial, subjective flow quantities that may skew correlation values; that is, we simply skip the data pair (flow and climate index) in those missing-data years for the calculation of correlation values.

JAS streamflow data prepared thus far for the target upstream and downstream gauges represent potential predictands for seasonal forecasting, and this study plans to analyse how these predictands correlate with the major teleconnection patterns.
This correlation analysis is purposeful since the major teleconnection patterns can emulate a miniature of the full-scale climate system for a great deal of climate variability explained, thereby providing some clues regarding the causations of the interannual variations of Taiwan's streamflow. A list of the major teleconnection indices has been compiled for this purpose. The desired list should cover as many teleconnection patterns as possible, but those selected should show certain signs of connections to East Asian climate based on previous entries in the literature. The ENSO is characterized as an air-sea coupled phenomenon: a
zonal Sea Level Pressure (SLP) anomaly in the tropical Pacific (i.e., the Southern Oscillation) and a quasi-periodic Sea Surface Temperature (SST) warming/cooling in the tropical eastern Pacific (i.e., El Niño/La Niña). As the impact of ENSO on East Asian climate is well known, the list begins with three ENSO indices, namely NINO 1+2, NINO 3.4, and NINO 4, which represent the sources of influence from East, East-Central, and Central Tropical Pacific, respectively. In the immediate vicinity of the Pacific, the Indian Ocean has the IOD as the leading mode with evidence of steering East Asian Monsoons and other
weather systems (Guan and Yamagata, 2003), so the IOD index is selected. Over the farther side and higher latitude of the Pacific, the EP-NP and PNA patterns, found to be associated with the intensity and location of the Pacific jet stream (e.g., Yang et al., 2002), are included in the list. Although sprung from the Polar Regions, it has been indicated by more and more studies that the AO and AAO can affect climate variability in remote, subtropical regions (e.g., Wang and Fan, 2005; Choi et al., 2012); it would be reasonable to have these two indices in the list. To account for possible transoceanic interactions recently addressed
by various studies (Hong et al., 2014b), the NAO index, referred to as the meridional seesaw of the SLP field with the north and south centres near Iceland and the Azores, respectively, is also included in the list. Furthermore, as the predictand of interest is highly related to summertime tropical cyclone (TC) activity, the list contains the QBO (Quasi-Biennial Oscillation, Baldwin et al., 2001), WP, and PJ indices (Choi et al., 2010; Kosaka et al., 2013). The QBO depicts a quasi-periodic oscillation between easterlies (positive) and westerlies (negative phase) over the lower tropical stratosphere, and the period is approximately 20
to 36 months. The WP pattern is one of the leading modes of low-frequency variability over the North Pacific, consisting of three anomaly centres: a meridional dipole with one centre near the Kamchatka Peninsula and another over the subtropical western North Pacific (WNP) and a third centre over the eastern North Pacific. The PJ pattern is a meridional teleconnection characterized by cyclonic and anticyclonic anomalies over the midlatitude WNP. Beyond the above climate indices, the PDO and AMO indices, as representatives of the interdecadal oscillations, are also included in the list for the examination of any
low-frequency connections. The PDO is characterized by a long-lived ENSO-like pattern that shifts phases with a period of at least 15 to 25 years, and the AMO is signified with long-term SST variations over the North Atlantic Ocean with a period of 50 to 70 years. Table 2 displays the list of the climate indices and depicts their sources as well as some key references.

To reveal large-scale patterns pertaining to certain climate indices (or even unprecedented signals) that potentially dominate Taiwan's streamflow variability, this study uses various other data sets, including the Extended Reconstructed Sea Surface





Temperature (ERSST Version 4, Huang et al., 2015), NCEP/NCAR Reanalysis SLP, geopotential height, and zonal wind (Kalnay et al., 1996), and Global Precipitation Climatology Project (GPCP Version 2, Adler et al., 2003).

## 2.2 Lagged correlation analysis

The design of the correlation analysis is explained as follows. First, correlation coefficients (Pearson's $r$ is used) between the

JAS flow data at one of the gauges and different climate indices in the same season (e.g., the JAS ENSO index) are calculated. The calculation of the "concurrent" correlations is then repeated until all the gauges are covered. Second, because the motivation of this work is to explore any forecasting possibilities, lagged correlations are computed as well. Lagged correlations are calculated between the JAS flow data and the climate indices averaged over the preceding three-, six-, and nine-month seasons, namely, AMJ, JFMAMJ, and ONDJFMAMJ, respectively. This approach is commonly adopted by plentiful forecasting

studies (e.g., Rajeevan et al., 2007) as the extension of averaged periods can eliminate some high-frequency or artificial data disturbances. To examine prediction skills for longer lead times, lagged correlations between the JAS flow data and the climate indices averaged over two preceding periods, ONDJFM and the previous OND, are calculated as well. The above correlation analysis is applied to all of the climate indices. Note that the PJ index supplied by the source (Kubota et al., 2016) is derived from the JJA data only, so we have used the following steps to develop the new PJ index (with Dr. Kubota's guidance) to pursue

consistent analysis with the remaining indices: 1) we first obtained atmospheric pressure data at Yokohama and Hengchun in Japan and Taiwan, respectively; 2) we calculated the JAS (and all other tri-monthly periods, e.g., AMJ, JFM, and the previous OND) average of the atmospheric pressure anomaly and then normalized the values by the standard deviation at each station; 3) the JAS (and all other tri-monthly periods) PJ index was derived from the difference of the two normalized pressure anomalies, and then the index was normalized again by its 1979–2009 standard deviation. Using the above steps, we have successfully

reproduced the JJA PJ index and produced the new JAS PJ index. Figure S1 (supplementary material) shows the comparison between the PJ index in JAS and that in JJA.

It should be noted that the concurrent analysis does not produce immediate forecasting utility. However, we believe that it is still important to examine concurrent relationships between climate indices and streamflow since many climate patterns have been proven to drive regional climates in the current season. The idea of calculating contemporaneous correlations was

likely best demonstrated by Wallace and Gutzler (1981), who nicely described several dominant teleconnection patterns at the Northern Hemisphere extratropics during winter (e.g., NAO). Beyond the Northern Hemisphere extratropics, one of the most important concurrent relationships witnessed by several operating agencies and research organizations (e.g., CPC and IRI) is the impact of ENSO on world regions. Various maps of the concurrent relationships (e.g., composite and historical probability) have been archived as valuable references. Over the Indian Ocean basin, the different phases of the IOD are also known to

have pronounced concurrent impacts on the formation of the trade wind and the short rains over East Africa from October to November (Black et al., 2003; Clark et al., 2003; Behera et al., 2005; Chen and Georgakakos, 2015). In contrast, significant lagged correlations (if identified) can indeed generate some forecasting utility, but to assess the dynamical mechanisms of the lagged relationships found by statistical approaches is usually not a trivial task. To use concurrent relationships for forecasting,





one can adopt a hierarchical or hybrid approach that applies another empirical or dynamical model to forecast the climate indices in the first instance (e.g., Kim and Webster, 2010).

One of the most fundamental assumptions of the correlation analysis is that the result of such analysis does not indicate any causality. The result can be two-way; that is, there is no physical implication for a predictor-predictand relationship. However,

the assumption held by us is that significant correlations should suggest some large-scale dominance over the local-scale hydroclimate since the opposite route of dominance (i.e., the impact of a disturbance at island (Taiwan)- scale on large-scale circulations) is unlikely and hard to explain. In addition, we also neglect the effect of any outliers (if they exist) and examine only the linear relationship between two continuous variables. In other words, our analysis cannot identify any nonlinear effect of extreme teleconnection patterns on Taiwan's streamflow.

## 2.3 Climate regime shift test

As stated in the study objective, following correlation analysis is the examination of the temporal disruption of found significant correlations. The temporal disruption or abrupt changes in a univariate time series can be commonly identified by using classic, nonparametric techniques, such as the Mann-Whitney-Pettitt (MWP, Pettitt, 1979) and Kruskal-Wallis (KW, Kruskal and Wallis, 1952) tests. However, these techniques may not be directly applicable to quantities such as temporal correlations

derived from a bivariate time series. A simple resolution of the change-point identification under this circumstance would be the usage of the moving-window approach to obtain a "correlation time series" to which either the MWP or KW test can be applied. Another statistically sound approach to this problem, proposed by Rodionov (2015), is used in this study. Rodionov's method in detecting abrupt changes in the correlation coefficient is based on the fundamental property of variance. If there are two variables of interest, $x$ and $y$ (e.g., the streamflow and climate index), the variance of the sum of $x$ and $y$ can be written as:

$$S^2_{x+y} = S^2_x + S^2_y + 2rS_xSy, \tag{1}$$

where $S^2$, $S$, and $r$ denote the sample variance, standard deviation, and correlation coefficient, respectively. Further, if the two variables have zero mean and unit variance, the above equation can be reduced to:

$$S^2_{x+y} = 2(1+r). \tag{2}$$

Note that, because the sample correlation coefficient ranges from -1 to 1, the above variance is bounded between 0 and 4. Equation 2 also indicates that the identification of shifts in $r$ is equivalent to that in $S^2_{x+y}$. In his previous work, Rodionov (2005) introduced a method for detecting the abrupt shifts in the variance (of a single variable) based on a "sequential $F$-test." Therefore, the same method for the variance of $x$ or $y$ can be simply applied to the variance of $x + y$, thereby achieving the identification of shifts in the correlation coefficient. In essence, the above method can also be applied to the variance of $x - y$,

$$S^2_{x-y} = 2(1-r), \tag{3}$$





which should theoretically yield very similar change-points if the $p$-value (computed from the Fisher's $r$-to-$z$-transformation, Fisher, 1921) is less than 0.05 (i.e., a high significance level). As per Rodionov's suggestion, the test is performed for both the sum and difference series to ensure that the minimum $p$-value is attained.

If the variables have not been normalized, Rodionov (2015) stated that some pre-processing work on the time series is
required: using the shift detection in the mean (based on a "sequential $t$-test," Rodionov, 2004) and variance to obtain the stepwise means/trends and variances, respectively. The variables can then be normalized for the shift detection in the correlation coefficient. In addition to the pre-processing work and the desired significance level, a cut-off length $l$ should be determined for the method to detect change-points and to calculate associated statistics. For more discussion regarding the caveats of using the CRS detection method and its detailed documentation, please refer to the supplementary material and Rodionov's previous
studies.

## 2.4   Linear regression prediction model

Linear regression is widely used in climate forecasting studies (e.g., Hastenrath, 1995; Hastenrath et al., 2004; Chen and Georgakakos, 2015) to generate forecasts based on a calibrated, linear equation that depicts how a hydro-climatic predictand ($\boldsymbol{Y}$, the streamflow in our case) responds to selected predictors ($\boldsymbol{X}$, climate indices in our case):

$$\boldsymbol{Y} = \boldsymbol{X}\beta, \tag{4}$$

where $\beta$ represents coefficients estimated by ordinary least squares. In each catchment, we develop a linear regression equation as the prediction model. In terms of predictors (i.e., independent variables), we adopt 13 climate indices described in the paper (with AAO excluded as relatively short in record) and perform stepwise model selection based on AIC (Akaike Information Criteria). Model selection can be performed in the forward or backward direction, and we use both directions to ensure a thor-
ough search in the variable space. Afterwards, to avoid possible multicollinearity issues resulting from some highly correlated climate indices, the variance inflation factor ($VIF$) is assessed:

$$VIF_j = \frac{1}{1 - R_j^2}, \tag{5}$$

where $R_j^2$ is the coefficient of determination from a regression of the $j^{th}$ predictor on any other predictors. According to the literature (Chen and Georgakakos, 2014; Hidalgo-Muñoz et al., 2015), the $VIF$ tolerance threshold is set to be 4 for small
samples (say $\sim 50$ points). The final model is thus determined and used to generate hindcasts (i.e., retrospective forecasts) for that catchment. The generation of hindcasts is subject to the leave-one-out cross-validation (LOOCV) procedure to circumvent artificial skill. Lastly, the LOOCV correlation and Gerrity Skill Score (GSS Gerrity, 1992) are calculated to assess the prediction skill in that catchment. The GSS, unlike other conventional skill scores (e.g., the Heidke skill score), is an "equitable skill score" with a specific scoring matrix known for rewarding correct forecasts in the less ordinary categories (e.g., high and low flows
in our case). Alternatively, the GSS does not reward random or constant forecasts. The GSS has been adopted by a number of practitioners of climate forecasts (e.g., CPC and WMO) and many studies (e.g., Chen and Georgakakos, 2014; Hidalgo-Muñoz



et al., 2015) as a standard multi-categorical skill score, and it is thus used in this study. The above framework is repeated for all 41 catchments.

## 3 Results

### 3.1 Correlations between Taiwan's runoff and climate indices

In line with the aforementioned instruction, the correlation analysis is conducted for all the target gauges in Taiwan. Because the total number of combinations of the different gauges (upstream and downstream), climate indices, and lagged periods is in the thousands, the resulting correlation values are merely too many to be fully listed here. Therefore, the results are presented in a selective and illustrative fashion. In Figures 2 and 3, concurrent and lagged correlations between the JAS runoff at the upstream gauges and selected climate indices are colour-coded over the maps of the catchments. The criterion for selecting

climate indices is that they must exhibit correlation values at the 95% confidence level with at least one of the catchments under either the concurrent or lagged scenario. In other words, those climate indices excluded from the plots do not show significant concurrent or lagged correlations with any of the upstream catchments. In addition, please note that the lagged correlations shown in Figure 3 are based on the average period over ONDJFMAMJ only (lagged correlations based on other average periods, see Section 2.2, show quite similar patterns and are available upon request). In the two figures, the highest

absolute correlation value among all the upstream catchments for a specific climate index is also denoted in each plot.

    Several key messages can be deciphered from Figures 2 and 3. Under the concurrent scenario, 1) many upstream catchments show significant positive correlations with the climate indices, and among these indices, the WP and PJ indices stand out as having the strongest, universally in-phase relationship with the JAS runoff, indicating the direct influence of typhoon activity during the same season; 2) the only climate index showing all negative correlations with the upstream catchments is the PDO,

and this out-of-phase relationship is supported by some existing findings (e.g., Li et al., 2010) regarding the PDO's influence on East Asian climates; and 3) surprisingly, the NAO, as the farthest teleconnection mode, shows quite strong correlations (∼99% confidence level) with some upstream catchments. Alternatively, under the lagged scenario, 1) in comparison with the concurrent correlation patterns, some of them experience a clear phase reversal over certain regions of Taiwan (e.g., North-Northeast for IOD and WP and Central-Southwest for PNA); and 2) the most pronounced out-of-phase relationship is observed

from the QBO index, and this relationship is even stronger than any of the concurrent relationships.

    To further interpret the range of correlation values, Figure 4 encapsulates the correlations of the JAS runoff at every gauge against every climate index using box-and-whisker plots. Each box plot constituted by 28 (13) values represents the range of correlations observed from the upstream (downstream) gauges. The results for upstream and downstream gauges are separately illustrated in the left and right panels, respectively. Concurrent and lagged correlations with JAS, ONDJFMAMJ, ONDJFM,

and OND climate indices are shown from the top to bottom of Figure 4 in sequence. Moreover, the abscissa of Figures 4a and 4b is ranked by the mean correlation in each box plot for a clearer illustration of the performance between climate indices. To observe whether a phase transition occurs, the same ranking is then inherited by Figures 4c through 4h. In addition, the





corresponding correlation values for all the downstream gauges are also enumerated in Tables 3 and 4 (values for the upstream gauges are available upon request).

Compare Figure 4a with 4b: the ranking of the climate index is nearly the same (except PNA), and the interquartile range (IQR) of each box plot is also similar, implying the general scale (of catchments) consistency in response to large-scale circulations and the small influence of human intervention on the JAS runoff for the downstream. However, the total ranges of correlation values are wider for the upstream, which possibly reflects the higher randomness of catchments at a smaller scale. Under the concurrent scenario, out of the 14 climate indices, ten and nine tend to positively correlate with the JAS runoff for the upstream and downstream, respectively. While under the lagged scenario (Figures 4c and 4d), only four climate indices are positively correlated with the JAS runoff for both the upstream and downstream gauges (i.e., phase reversal mentioned earlier). In fact, the PNA, IOD, NINO4, NINO3.4, and WP are the climate indices showing the phase reversal for both the upstream and downstream JAS runoff. Among all the indices, the QBO again indicates the strongest negative correlations with Taiwan's streamflow as the lead time increases.

To confirm the best outstanding lagged correlation between Taiwan's streamflow and the QBO, additional literature reviews and field significance tests still need to be conducted. The strongest lagged correlation is very likely attributed to the TC activity in the WNP modulated by the QBO. Chan (2003) has performed a cross-spectral analysis between the QBO and the number of TCs in the WNP and indicated that the leading westerly phase of the QBO can result in an increase in TC activity. He explained that the westerly phase of the QBO creates an environment of relatively low vertical wind shear in favour of TC formation. Ho et al. (2009) later found that, during the westerly (easterly) phase of the QBO, more TCs approach the East China Sea (the eastern shore of Japan). Therefore, the negative correlation between the QBO index and TC activity in the vicinity of Taiwan is carried over into the negative correlation with streamflow. In fact, such strong correlations found in 22 out of the 41 catchments also reach field significance. The number of catchments with significant temporal correlations has exceeded the critical value of field significance ($p = 0.05$) from the empirical null distribution (Figure S2, supplementary material) developed by using a Monte Carlo technique similar to those suggested by Livezey and Chen (1983) and Wilks (2011). 2000 Monte Carlo trials are used, and each trial depicts a significant local test for correlations between the "randomly ordered" QBO index and streamflow data at the 41 catchments, resulting in a count of the number of catchments with significant temporal correlations constituting the null distribution.

## 3.2   Regime shifts in the correlation

Although some of the correlation results for certain catchments against climate indices disclosed above are at a moderate significance level, it is found that these correlations are subject to peculiar interannual variations. For instance, 20-year moving-window correlations with four climate indices, namely the PDO, EPNP, AO, and PJ, are examined (Figure S3, supplementary material). From those plots in Figure S3, several types of interannual variations as potential signs of CRS can be identified: 1) divergence of the total ranges over time (e.g., Plots a, b, and c), 2) a gradual decreasing trend (e.g., Plot b), 3) an abrupt increase or decrease (e.g., Plots c and d), and 4) a change of sign in the mean correlation near the suspicious CRS year (e.g., Plots b and c). We thus hypothesized that the relationship between Taiwan's streamflow and large-scale circulations may have undergone





several climate regime shifts. To validate this hypothesis and identify possible years of CRS, the CRS test is applied to the streamflow data and climate indices. Based on the spatial significance of the results of correlation analysis, the aggregated JAS runoff for East (West) Taiwan is computed from the average of HLI and SGL (WU, JS, and BG) data for the CRS test. Figure S4 (supplementary material) presents the result of the CRS test for identifying any shifts in the correlation between the East and

West Taiwan runoff versus the PJ index. The result indicates two highly significant change points in 1979 and 1999 (significant at $p = 0.0001$ and $0.0003$, respectively) for the East Taiwan runoff. Between 1979 and 1999, the correlation value is higher than 0.8, suggesting the dominant effect of the PJ pattern on the moisture transport to Taiwan in summer. Such strong correlation is hardly ever seen in the field of climate sciences; however, the correlation deteriorates drastically after 1999, becoming slightly negative and close to zero. Likewise, two very significant change points in 1988 and 2000 (significant at $p = 0.006$ and $0.0003$,

respectively) can be identified for the West Taiwan runoff. While the first change point is identified in the late 1980s, the marked in-phase relationship between 1979 and 1999 is still quite notable. The same CRS test has been applied to Taiwan's runoff versus all the other climate indices, but only less significant results can be found. In the next subsection, we will put the above correlation and CRS analyses in the context of prediction through a hindcasting experiment.

### 3.3   Variations in prediction skill due to the shifts

Figure 5 shows some hindcasting results for selected upstream and downstream catchments. LOOCV correlations vary from one catchment to another and can be as high as $\sim 0.6$. As a more stringent metric, cross-validated GSS values are generally lower but most of the time pass the significant threshold (e.g., $\sim 0.25$ for data size 30, determined by the bootstrap-based hypothesis testing, Chen and Georgakakos, 2014). Overall, using large-scale circulation indices can produce fair to good prediction skills in summer streamflow prediction in Taiwan. Among those many climate indices, the PDO and PJ indices are

selected most frequently as the predictors for the catchments [while the former is selected seven times (except for SGL), the latter is selected five times (except for Catchments 3 and 9 and BG) for the results shown in Figure 5]. This result is, to a certain extent, consistent with our general correlation assessments (e.g., Table 3) and indicates the general dominance of summer climate in Taiwan. Regarding the origin of the predictability, Chen and Chen (2011) indicated that the PDO coincides with the specific meridional SST contrast (i.e., warming in the tropical central and eastern Pacific and cooling in the extratropical North

Pacific), which plays a dominant role in modulating summer rainfall in Taiwan. Choi et al. (2010), Kosaka et al. (2013), and Kubota et al. (2016) all provided sufficient evidence of the significant impact of the PJ on tropical cyclone activity and rainfall over the WNP during summer. Based on our findings, the predictability for summer rainfall can be extended to streamflow in Taiwan.

From Figure 5, we can note that the relatively better performance of each LOOCV time series seems to occur during the

period from the late 1970s to the late 1990s, coinciding with the CRS epoch discussed earlier. Hindcasts during the pre-regime shift epoch seem to still be able to capture the general variability of the observed runoff (with relatively poorer performance), whereas hindcasts during the post-regime shift epoch appear to present more opposite signals and apparent departures from the observed runoff. It is worth noting that some of the departures occur in years when JAS typhoon activity is abnormally high. For example, in 2007, the typhoons Pabuk, Sepat, and Wipha together generated the highest amount of cumulative rainfall





for some watersheds over the past decade, and in 2008, the typhoons Kalmaegi, Fung-Wong, Sinlaku, Hagupit, and Jangmi made a record of continuous invasions of intense typhoons (all Category 2 and above) in JAS. To further illustrate the effect of CRSs on streamflow prediction, we fitted a new regression model using the data from 1979 to 1998 and then evaluated how the fitted model performed in the remaining years. Using the SGL watershed as an example, Figure 5i shows the new

hindcasting result. In comparison with Figure 5h, the new fitted model exhibits some definite improvement during the period from 1979 to 1998, showing the outstanding CV correlation and GSS values as 0.84 and 0.56, respectively. However, the fitted model can generate nothing but extremely poor hindcasts for the remainders. In fact, both skill metrics show a reverse sign, clearly illustrating distinct climate regimes over the temporal horizon. In contrast with the above experiment, if we fit another regression model using the data outside the time frame of 1979–98, the stepwise model selection scheme discloses no climate

indices that are qualified to be a predictor. To summarize, the linear regression experiment indicates that the assumption of a stable predictor-predictand relationship could be quite problematic for hydro-climatic forecasting due to the observation of CRS.

## 4    Discussion

In this section, further discussion is provided to address the issue of CRS and how it can impact the convention of seasonal

forecasting evidenced by some large-scale patterns. To start the discussion, we would like to argue that the two change points found for the response of Taiwan's streamflow to large-scale circulations are not a coincidence. Whereas the first change point in the late 1970s (for the East Taiwan runoff in particular) is clearly in relation to the widely known CRS identified over the entire Pacific basin SST (Miller et al., 1994; Hare and Mantua, 2000), the second change point in the late 1990s coincides with the CRS induced by a warming over the equatorial western Pacific (McPhaden et al., 2011; Hong et al., 2014a) and/or more

frequent occurrences of the central Pacific El Niño (Xiang et al., 2009). To supplement our explanation here, the CRS analysis of shifts in the mean is also applied to all the climate indices examined in this study, and the change points corresponding to each climate index are listed in Table 5. From the table, the shifts in the mean identified for the PDO are very much consistent with previous studies, and only such shifts (rather than the PJ) are in perfect agreement with the identified shifts in the correlation for Taiwan's streamflow. Therefore, our hypothesis regarding the identified CRS can be drawn as follows:

the CRS first emanates from the change in the basin-scale climatology over the Pacific (e.g., shift in the PDO), and then the reorganized large-scale patterns can reset the relationship between the island-scale streamflow with established regional circulations (e.g., the PJ pattern).

     The above hypothesis seems to be supported by some existing findings. Because the PDO is strongly tied to ENSO, the shift in the PDO can induce changes in the ENSO-related SST anomalies as well as ENSO-related teleconnections (Duan et al.,

2013). Such SST anomalies have been found to be robust in the WNP during summer (Alexander et al., 2002), and numerical model experiments have verified the ENSO-forced PJ pattern (Kosaka et al., 2013). Kubota et al. (2016) further noted that the ENSO-PJ relationship was strengthened after 1980 and then weakened after 2000, likely due to the phase shift in the PDO. When the ENSO-PJ relationship is more pronounced, the systematic impacts of the PJ pattern on TC activity, rainfall, and,





subsequently, streamflow in Taiwan are clear. By contrast, if the PJ pattern is less forced by ENSO, the associated impacts can be ambiguous. Consequently, Taiwan's streamflow become less predictable after the post-regime shift epoch.

The observation of the CRS can profoundly influence predictor screening for empirical forecasting methods. Conventional predictor screening usually relies on pattern correlation (Chen and Georgakakos, 2014), that is, identifying specific areas over a predictor field (usually SST for its considerable energy absorption and slowly varying property) showing significant correlation with the predictand of interest (e.g., local precipitation or streamflow) over a sufficiently long period of time. This concept holds true if the predictor-predictand relationship remains quasi-stationary. In contrast, if the predictor-predictand relationship is no longer stationary (e.g., identification of a prominent CRS in the correlation), this concept of predictor screening would become rather questionable. To illustrate the impact of the CRS on seasonal forecasting, the PJ pattern, which shows the overall highest concurrent correlation with the JAS runoff in Taiwan over a specific time window, is used to generate some large-scale patterns next.

First, two sets of pattern correlation of the PJ index with the SLP, SST, or GPCP data are generated using 1999 as a demarcation of the time window (Figure 6). Before 1999, the PJ-SLP correlation indicates a marked dipole pattern with its two poles centred at southern Japan and near Taiwan. Along with the significant negative correlation (at $p = 0.05$) extended to the Indian Ocean (regarding SST) and the positive correlation over Taiwan (regarding GPCP), all these patterns reasonably reproduce the canonical PJ pattern defined by previous studies (e.g., Kubota et al., 2016). In contrast, from 1999 onward, the entire set of pattern correlation is altered dramatically: the dipole pattern for the PJ-SLP correlation becomes less significant, and the extensive negative correlation over the Indian Ocean for the PJ-SST correlation reverses sign. In addition, the positive correlation originally occupying the Taiwan area migrates northeast to the East China Sea. The above analysis implies that owing to a possible CRS, a specific climate index (e.g., PJ) can respond quite differently to large-scale circulations from one time window to another. More importantly, if that climate index is adopted as a predictor (often seen in numerous long-lead forecasting applications, e.g., Moradkhani and Meier, 2010; Robertson and Wang, 2012; Hidalgo-Muñoz et al., 2015), its relationship with the predictand (e.g., precipitation over Taiwan) could largely weaken or even be terminated given the existence of a CRS.

If pattern correlation is calculated directly between the East Taiwan runoff (as the predictand) and the SLP, SST, or GPCP data (as the predictor fields), the bulk of significant areas in the predictor fields usually specifies the spatial extent of potential predictors, in accordance with the common procedure of predictor screening appearing in many articles (DelSole and Shukla, 2009). Figure 7 demonstrates this procedure twice, once before and once after the identified CRS. Obviously, the left column of Figure 7 resembles that of Figure 6 to a very high degree, indicating that the PJ pattern is indeed an ideal predictor candidate before 1999. Nevertheless, from 1999 onward, the right column of Figure 7 resembles neither the left column of the same figure nor the right column of Figure 6, implying that the PJ predictor should be replaced by some other predictor candidates (e.g., SST over the subtropical central Pacific).

Lastly, to achieve some forecasting utility, lead time information can be incorporated into the procedure of predictor screening. For instance, Figure 8 presents the evolution of the SST anomaly (SSTA) composites based on wet-minus-dry years of the JAS runoff for East Taiwan, which can also reveal significant predictor areas at varied lead times (Chen and Georgakakos, 2015). The identified CRS is used to divide the generation of the SSTA composites. Before the CRS, (1982, 1984, 1985, and



1994) and (1980, 1983, 1993, and 1998) are identified as the wet and dry years, respectively. After the CRS, (2001, 2005, and 2007) and (1999, 2002, and 2010) are identified as the wet and dry years, respectively. Wet and dry years are identified as those years for which the JAS runoff is higher and lower than the $67^{th}$ and $33^{rd}$ percentiles of the entire data series, respectively. During the concurrent season, notwithstanding that Figures 8(a) and (b) show the SSTA patterns as extensions of the patterns

observed in Figures 7(c) and (d), it can be found that the change in the patterns before and after the CRS is significant up to the global scale. With increases in lead times, the two sets of SSTA composites present distinct evolutionary pathways. Eventually, in previous OND, the SSTA composite expresses a La Niña pattern for the before-CRS scenario, completely opposite of the El Niño pattern for the after-CRS scenario. No significant SST difference relevant to wet or dry can be identified after the CRS, indicating the weakened source of predictability from the SST.

It would be worthwhile to examine the evolution of composites for other variables, especially for those used to define such potential predictors as the WP, PJ, and QBO patterns. According to Table 2, these variables represent 500-mb geopotential height, sea level pressure, and 30-mb zonal wind. Therefore, the wet and dry contrast of these variables, before and after the CRS, are presented in Figures 9 and 10 using the same format as Figure 8 (geopotential height plots are not shown, as they are quite similar to the sea level pressure ones). From the sea level pressure composites, before the CRS, significant cyclonic

anomalies over the WNP can be found from the concurrent typhoon season to even earlier periods of the same year [Figures 9(a), (c), and (e)]. After the CRS, while the WNP area is inactive with no significant cyclonic anomalies, some patterns with signs opposite of those before the CRS can be found in the mid-to-high latitude regions at varied lead times. From the zonal wind composites shown in Figure 10, the phase reversal is markedly along the equator: whereas the zonal wind pattern evolves into (backward in time) clear easterly anomalies under the before-CRS scenario, strong easterly anomalies change into westerly

anomalies in previous OND under the after-CRS scenario. Nevertheless, it should be noted that no statistically significant patterns for the wind anomalies are identified along the equator, suggesting that further analysis should be conducted to validate the predictor-predictand relationship between the QBO and Taiwan's streamflow. Overall, the SSTA, sea level pressure, and zonal wind composites all consistently indicate that the large-scale patterns reverse phases. Our discussion herein illustrates the importance of a predictor screening scheme to account for a forthcoming CRS.

**5   Summary and Conclusion**

A set of teleconnection patterns is a proxy for the complex climate system, and these patterns have shown dominant effects on modulating regional hydro-climates in different seasons. A response of a region's hydro-climates to the global climate system can be characterized by the regional sink of moisture transported via various circulations and mechanisms. This study has attempted to shed light on the relationship between teleconnection patterns and Taiwan's streamflow. It is believed that

streamflow data could be a more descriptive and preferable metric than precipitation relating to teleconnection patterns for its sustained (i.e., less transient) integrated water response to the climate system and direct association with water resources and hydro-meteorological hazards. Long-term JAS streamflow data derived at 28 upstream and 13 downstream gauges in Taiwan were used to correlate with 14 teleconnection indices showing signs of linkage to East Asian climate. To scrutinize the potential





forecasting utility, the climate indices were averaged over not only JAS but also the preceding seasons for the calculation of concurrent and lagged correlations. In the course of our correlation analysis, it was noted that some significant correlation results based on the entire period of record were actually induced by an even more significant in-phase (or out-of-phase) relationship during a truncated time frame, indicating inherent climate regime shifts. Therefore, CRS analysis was performed

to identify any significant shifts in the correlation, as well as the mean of the climate indices themselves. Discussion of how the identified regime shifts impact empirical prediction was then carried out. Our key findings are summarized below.

1. Among those many teleconnection patterns, we found that the WP and PJ patterns exhibit the highest concurrent correlations with the JAS flows at both the upstream and downstream catchments in Taiwan. The determinant of such correlation performance should be the association of these patterns with cyclonic activity in the western North Pacific basin.

2. Alternatively, the lagged correlation analysis indicated that the QBO index is significantly correlated with the JAS flows in Taiwan, promoting its forecasting utility. In addition, some of the climate indices change their relationships with the JAS flow from mostly positive concurrent correlations to negative lagged correlations.

3. We identified two shifts in the correlation between the PJ index and Taiwan's streamflow in the late 1970s and 1990s and then verified that these shifts should be bonded to some existing findings of the alteration of large-scale circulations in

the Pacific around the same time.

4. Pattern correlation and composite analysis further illustrated drastic changes of large-scale patterns before and after the change point in the late 1990s. Our results overturn the convention of predictor screening widely used for empirical forecasting, as the alignment of predictors can vary in consonance with certain interannual and interdecadal oscillations.

Our current endeavour includes applying a similar analysis framework to Taiwan's streamflow in other seasons. Once clear

evidence of climate regime shifts in the succeeding work can be gathered, a new predictor screening algorithm capable of accounting for CRSs will be developed. This algorithm will be incorporated into a seasonal streamflow forecasting model to improve water resource planning and management in East Asia.

*Author contributions.* C-J. and T-Y. designed and conducted the analysis and co-wrote the manuscript.

*Acknowledgements.* Work by C-J. Chen was supported by Taiwan's Ministry of Science and Technology under grant: MOST 104-2625-M-
005-007-MY2. Work by T-Y. Lee was also supported by Taiwan's Ministry of Science and Technology under grant: MOST 104-2116-M-003-005. The authors express the upmost gratitude to Dr. Hisayuki Kubota for helping us develop the new PJ index and the Water Resources Agency and Taiwan Power Company for providing the streamflow data. The authors also acknowledge the anonymous referees for their insightful comments.



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





**Table 1.** Periods of data record and missing data percentages for all 41 catchments used for our analysis. Note that we use only JAS data in each year, and the missing data percentage is referred to as the percentage of years in which no JAS data is available.

| Catchment (downstream) | Period of Record | Missing Data % | Catchment (upstream) | Period of Record | Missing Data % | Catchment (upstream) | Period of Record | Missing Data % |
|---|---|---|---|---|---|---|---|---|
| TC | 1951–2013 | 0% | Cat01 | 1970–2013 | 2.3% | Cat15 | 1970–2013 | 2.3% |
| HLO | 1981–2013 | 0% | Cat02 | 1970–2006 | 0% | Cat16 | 1971–2013 | 0% |
| WU | 1966–2013 | 2.1% | Cat03 | 1970–2002 | 0% | Cat17 | 1970–2013 | 11.4% |
| JS | 1965–2009 | 0% | Cat04 | 1970–2002 | 0% | Cat18 | 1971–2013 | 2.3% |
| BG | 1949–2013 | 0% | Cat05 | 1971–2007 | 0% | Cat19 | 1970–2013 | 11.4% |
| ZW | 1960–2013 | 0% | Cat06 | 1972–2013 | 2.4% | Cat20 | 1970–2013 | 11.4% |
| ER | 1971–2013 | 0% | Cat07 | 1970–2008 | 0% | Cat21 | 1970–2013 | 11.4% |
| GP | 1951–2010 | 0% | Cat08 | 1976–2013 | 5.3% | Cat22 | 1970–2001 | 0% |
| BN | 1948–2013 | 4.5% | Cat09 | 1972–2013 | 0% | Cat23 | 1974–2013 | 5% |
| SGL | 1969–2013 | 0% | Cat10 | 1970–2013 | 0% | Cat24 | 1977–2011 | 5.7% |
| HLI | 1969–2013 | 0% | Cat11 | 1970–2013 | 2.3% | Cat25 | 1977–2011 | 2.9% |
| HP | 1975–2013 | 5.1% | Cat12 | 1970–2008 | 0% | Cat26 | 1970–2012 | 2.3% |
| LY | 1949–2009 | 0% | Cat13 | 1970–2013 | 9.1% | Cat27 | 1970–2013 | 0% |
| | | | Cat14 | 1970–2013 | 4.5% | Cat28 | 1970–2013 | 2.3% |



**Table 2.** List of the 14 climate indices used in this study.

| Climate Index | Full Form (Variable†) | Source | Sample Reference |
|---|---|---|---|
| AMO | Atlantic Multidecadal Oscillation (SST) | NOAA/ESRL/PSD | Enfield et al. (2001) |
| PDO | Pacific Decadal Oscillation (SST) | jisao.washington.edu | Mantua et al. (1997) |
| NINO1+2 | ENSO index at East Equatorial Pacific (SST) | NOAA/NCEP/CPC | Trenberth (1997) |
| NINO3.4 | ENSO index at East-Central Pacific (SST) | NOAA/NCEP/CPC | Trenberth (1997) |
| NINO4 | ENSO index at Central Pacific (SST) | NOAA/NCEP/CPC | Trenberth and Stepaniak (2001) |
| IOD | Indian Ocean Dipole (SST) | jamstec.go.jp | Saji et al. (1999) |
| EPNP | East Pacific-North Pacific (500-mb HGT) | NOAA/ESRL/PSD | Barnston and Livezey (1987) |
| PNA | Pacific North America (500-mb HGT) | NOAA/ESRL/PSD | Wallace and Gutzler (1981) |
| AO | Artic Oscillation (1000-mb HGT) | NOAA/NCEP/CPC | Thompson and Wallace (1998) |
| AAO | Antarctic Oscillation (700-mb HGT) | NOAA/ESRL/PSD | Gong and Wang (1999) |
| NAO | North Atlantic Oscillation (500-mb HGT) | NOAA/ESRL/PSD | Hurrell (1995) |
| QBO | Quasi Biennial Oscillation (30-mb UWND) | NOAA/ESRL/PSD | Naujokat (1986) |
| WP | West Pacific (500-mb HGT) | NOAA/ESRL/PSD | Barnston and Livezey (1987) |
| PJ | Pacific Japan (SLP) | From Dr. Hisayuki Kubota | Kubota et al. (2016) |

†: SST, HGT, UWND, and SLP stand for sea surface temperature, geopotential height, zonal wind, and sea level pressure, respectively.



**Table 3.** Results of correlation analysis for the major watersheds in Taiwan; values before (after) the slash are concurrent (lagged, ONDJFMAMJ) correlation coefficients ($\times 10^{-2}$; significant at $p = 0.05$ are bold and italic).

| Watershed† | AMO | PDO | NINO1+2 | NINO3.4 | NINO4 | IOD | EPNP | PNA | AO | AAO | NAO | QBO | WP | PJ |
|---|---|---|---|---|---|---|---|---|---|---|---|---|---|---|
| TC (NW) | 1/0 | -10/1 | -11/-15 | 21/-9 | *25/1* | 2/-9 | -1/7 | -6/-12 | 4/0 | 3/-4 | *28/8* | 0/-20 | 10/-9 | *33/-8* |
| HLO (NW) | 29/24 | -28/-17 | -23/-32 | 13/-23 | 16/-11 | -7/-9 | -23/5 | 9/-20 | 5/-7 | 14/7 | 16/-5 | -2/*-34* | 26/-5 | *45/21* |
| WU (W) | -14/-24 | 4/0 | -5/-16 | 11/-11 | 6/-9 | 11/-10 | 15/2 | -12/-13 | 3/11 | 0/2 | 22/26 | -2/-5 | *37/0* | 26/12 |
| JS (W) | 17/8 | -22/-17 | -5/-27 | 8/-30 | 4/-25 | 11/-4 | -9/-19 | 24/-18 | -3/10 | 8/24 | 12/3 | -13/-28 | *36/-1* | *40/34* |
| BG (W) | -3/-7 | *-27/-6* | -12/-14 | 0/-13 | -6/-19 | 6/-8 | 11/-8 | -7/-4 | -7/-22 | 8/-3 | *25/-14* | 3/-21 | *31/13* | 13/8 |
| ZW (SW) | 8/3 | *-27/-28* | 4/-19 | 11/-19 | 5/-17 | 11/-6 | -5/-16 | 15/-16 | -3/7 | -3/-1 | 13/-7 | 1/*-27* | 19/-9 | 25/12 |
| ER (SW) | 9/15 | -22/-9 | -6/-15 | -6/-15 | -5/-12 | 18/3 | -24/-10 | 10/-4 | 7/5 | -15/10 | 12/6 | -19/-6 | 22/-9 | 12/3 |
| GP (SW) | -1/-2 | *-26/-14* | -9/-21 | 7/-20 | 9/-17 | 16/-5 | -2/-4 | 19/-15 | -4/10 | -8/15 | *26/8* | -3/-22 | *25/8* | *34/3* |
| BN (SE) | 18/16 | -10/-21 | -6/-24 | 15/-11 | 12/-7 | -1/7 | 5/-5 | 6/-11 | -7/-12 | -16/18 | -4/-15 | -5/-10 | 5/-15 | *25/-9* |
| SGL (E) | -5/-6 | -13/0 | -19/-25 | 10/-9 | 16/-3 | 3/-6 | 8/-7 | -8/-10 | 6/4 | 2/8 | 15/8 | -16/*-35* | 20/-17 | *40/11* |
| HLI (E) | 26/21 | *-36/-21* | -19/-27 | 7/-9 | 14/1 | 12/5 | -25/-14 | 17/-15 | 9/-8 | -2/11 | 7/-12 | -12/*-38* | 5/0 | *38/13* |
| HP (NE) | -9/7 | -7/-3 | -23/-17 | -3/2 | 5/10 | 0/2 | 0/-2 | 15/-18 | 8/11 | -26/-25 | 20/4 | -7/-30 | 20/-26 | 17/-2 |
| LY (NE) | 13/19 | -19/-6 | 2/-8 | 21/-5 | *26/-1* | 7/11 | -12/1 | 9/-1 | -5/-13 | -7/-11 | -10/-4 | -18/*-44* | -2/-4 | *31/-8* |

†: Inside the parenthesis is the relative location of each watershed in Taiwan (NW: NorthWest; W: West; SW: SouthWest; SE: SouthEast; E: East; NE: NorthEast).





**Table 4.** As in Table 3, but for lagged correlation analysis; values before (after) the slash are lagged correlation coefficients with preceding ONDJFM (OND) climate indices ($\times 10^{-2}$; significant at $p = 0.05$ are bold and italic).

| Watershed† | AMO | PDO | NINO1+2 | NINO3.4 | NINO4 | IOD | EPNP | PNA | AO | AAO | NAO | QBO | WP | PJ |
|---|---|---|---|---|---|---|---|---|---|---|---|---|---|---|
| TC (NW) | 4/5 | 4/7 | -13/-14 | -13/-14 | -1/-1 | -10/-14 | -1/12 | -9/-9 | -6/-1 | 12/0 | 1/6 | -20/-19 | -14/-9 | -11/-2 |
| HLO (NW) | 27/29 | -14/-10 | -32/-29 | -24/-23 | -12/-10 | -3/-5 | -27/-10 | -11/0 | -5/4 | 4/-15 | 2/12 | *-42/-43* | -4/0 | 21/20 |
| WU (W) | -22/-19 | 0/7 | -18/-17 | -14/-17 | -11/-18 | -14/-24 | -4/1 | -11/-2 | 7/8 | 6/-6 | 14/14 | -1/3 | -4/-3 | 10/4 |
| JS (W) | 8/15 | -16/-9 | -28/*-29* | *-32/-31* | -26/-26 | -13/-19 | -26/-11 | -15/8 | 8/12 | *30*/8 | 8/21 | -21/-17 | 9/6 | *43/34* |
| BG (W) | -6/-7 | 5/12 | -12/-12 | -12/-10 | -18/-18 | -7/-14 | -2/18 | 2/9 | *-28*/-13 | -9/-21 | -4/12 | -20/-17 | -1/-10 | 1/16 |
| ZW (SW) | 4/8 | -26/-19 | -25/*-28* | -22/-21 | -19/-17 | -12/-14 | -20/-8 | -16/-3 | 4/2 | 12/-11 | -5/3 | *-28*/-26 | 0/-9 | 21/23 |
| ER (SW) | 16/18 | -10/-7 | -16/-10 | -15/-13 | -11/-8 | -7/-13 | -14/0 | -5/0 | -1/-7 | 22/5 | 0/0 | -20/-17 | 8/4 | 21/21 |
| GP (SW) | 1/6 | -9/0 | -20/-21 | -21/-19 | -17/-17 | -9/-19 | -4/10 | -13/-5 | 5/2 | *28*/6 | 3/9 | 3/9 | 10/5 | 13/19 |
| BN (SE) | 16/*25* | -22/-24 | *-25*/-26 | -15/-17 | -9/-12 | 8/-5 | 2/-9 | -15/-20 | -13/-11 | 24/5 | 3/-7 | -9/-7 | -11/-15 | -7/-5 |
| SGL (E) | -5/-3 | 4/5 | -16/-7 | -8/-8 | -2/-1 | -4/2 | -11/-5 | -11/-8 | 0/-2 | 21/16 | 6/8 | *-31*/-24 | -8/-20 | 9/-7 |
| HLI (E) | 21/21 | -17/-16 | -22/-18 | -9/-7 | 1/2 | -2/0 | -26/-14 | -15/-3 | -12/-8 | 17/0 | -12/3 | *-34/-29* | 8/-13 | 20/9 |
| HP (NE) | 13/14 | 1/8 | -7/-4 | 3/5 | 12/13 | 7/9 | -1/13 | -20/-23 | 5/1 | -6/10 | -6/-1 | -28/-22 | -13/-30 | 11/1 |
| LY (NE) | 18/18 | -5/-5 | -11/-15 | -12/-14 | -5/-3 | 6/-5 | 7/6 | -4/-7 | -17/-17 | -1/-11 | 6/-5 | *-31/-25* | 1/-2 | 4/8 |

†: Inside the parenthesis is the relative location of each watershed in Taiwan (NW: NorthWest; W: West; SW: SouthWest; SE: SouthEast; E: East; NE: NorthEast).





**Table 5.** Change points (shifts in the mean) identified for all the JAS climate indices examined in this study; significance level is reported by the number of asterisks (*: $p \leq 0.05$; **: $p \leq 0.01$; ***: $p \leq 0.001$; ****: $p \leq 0.0001$).

| Climate Index | Change Points |
|:---:|:---:|
| AMO | 1962 (****), 1987 (*), 1998 (***) |
| PDO | 1976 (****), 1998 (**), 2010 (**) |
| NINO1+2 | None |
| NINO3.4 | None |
| NINO4 | 1990 (****) |
| IOD | 2006 (**) |
| EPNP | 1997 (****) |
| PNA | 2002 (**) |
| AO | 1982 (*) |
| AAO | None |
| NAO | 1967 (*), 2006 (*) |
| QBO | None |
| WP | 2002 (*) |
| PJ | None |





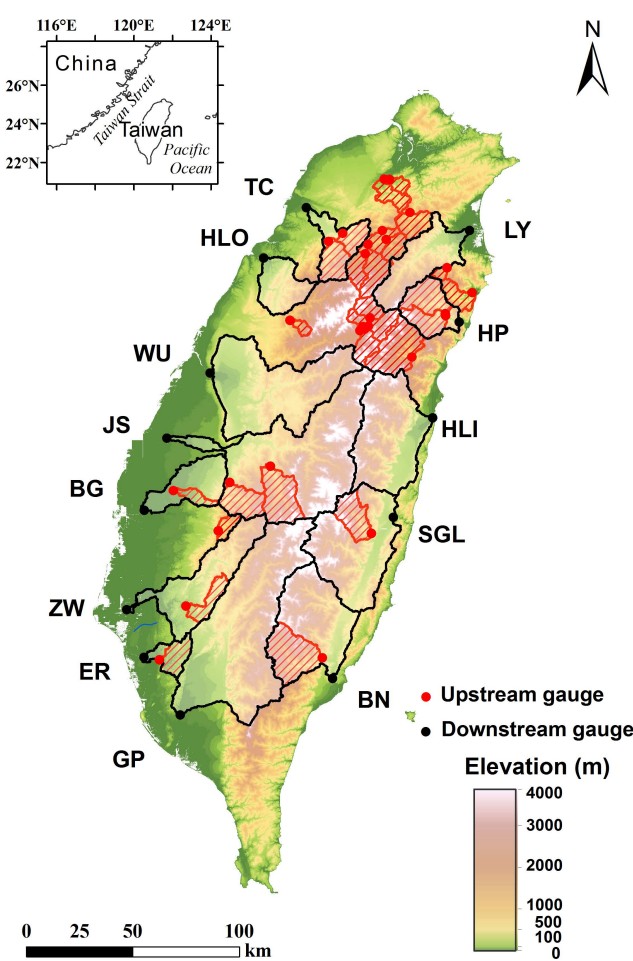

**Figure 1.** 13 major watersheds (black boundaries with abbreviations near the outlets) of Taiwan and 28 upstream catchments (red shadings) analysed in this study.





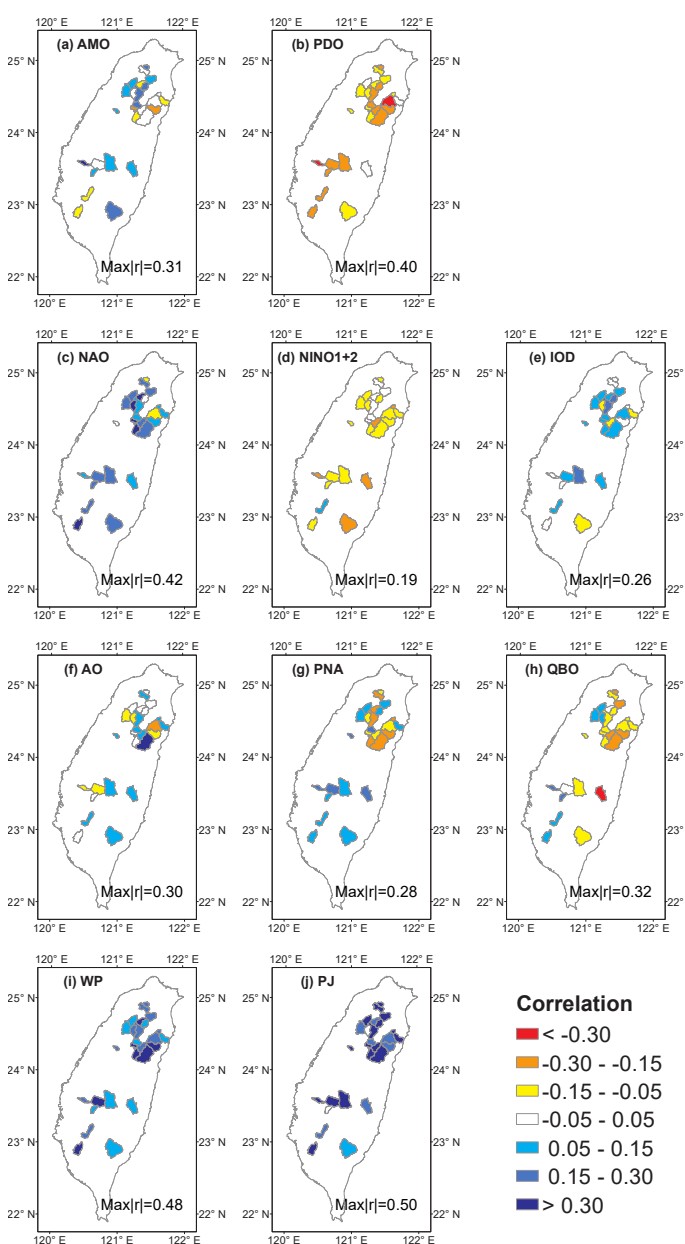

**Figure 2.** Maps of concurrent correlations showing each upstream catchment. The maximum absolute correlation value among all the catchments is denoted at the bottom of each plot.



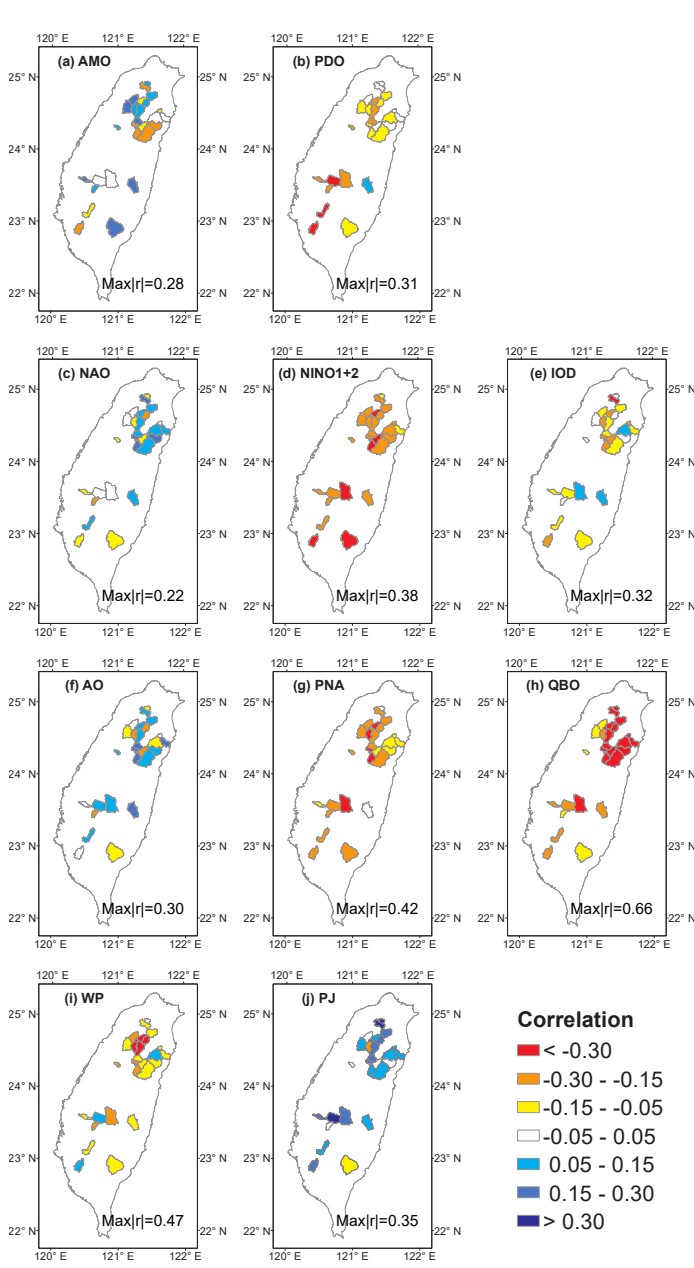

**Figure 3.** As in Figure 2, but for lagged correlations (JAS runoff vs. ONDJFMAMA climate indices).





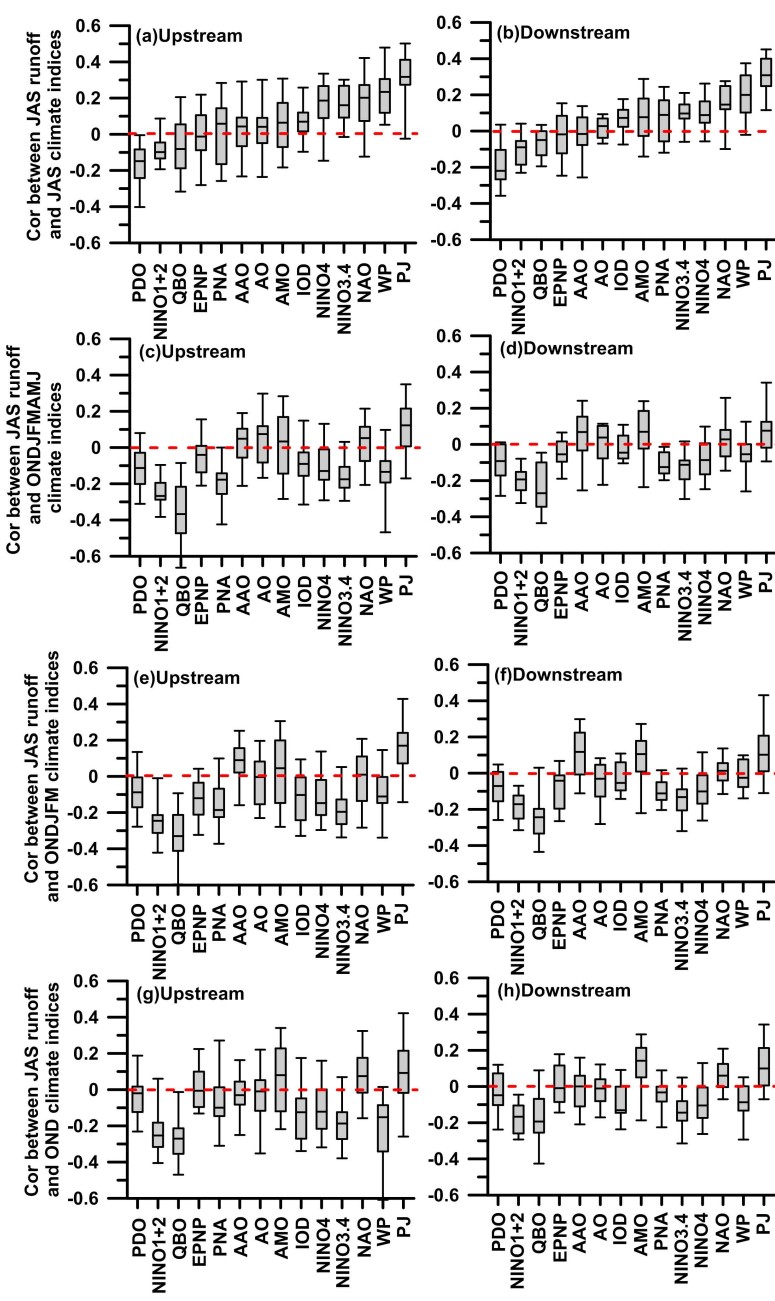

**Figure 4.** Box plots of correlation values of upstream (left) and downstream (right) runoff with climate indices. From top to bottom are concurrent to lagged correlations derived from climate indices averaged over different seasons.





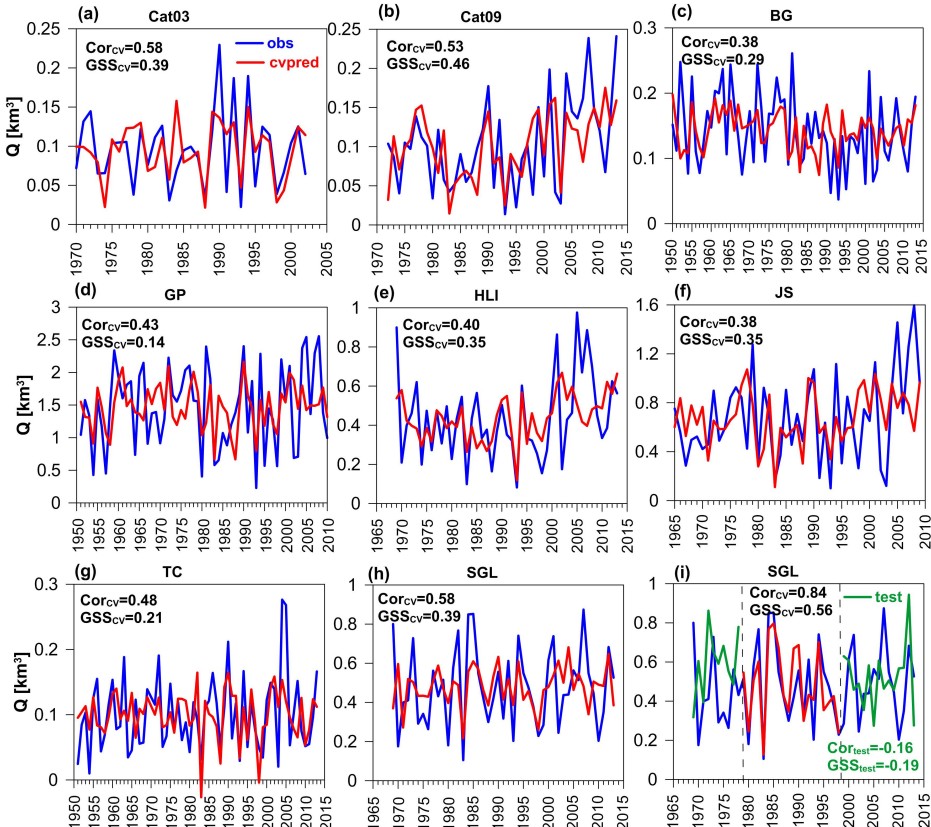

**Figure 5.** (a) to (h) are selected hindcasting results for upstream and downstream catchments in Taiwan using linear regression. Time series in red are model estimates based on the leave-one-out cross-validation (LOOCV) procedure. Cross-validated (CV) correlation and GSS values are also denoted in each plot. (i) is similar to (h), but the linear regression model is trained/fitted with the data from 1979 to 1998, and then the fitted model is tested with the rest of the data points (i.e., 1969–78 and 1999–2013)





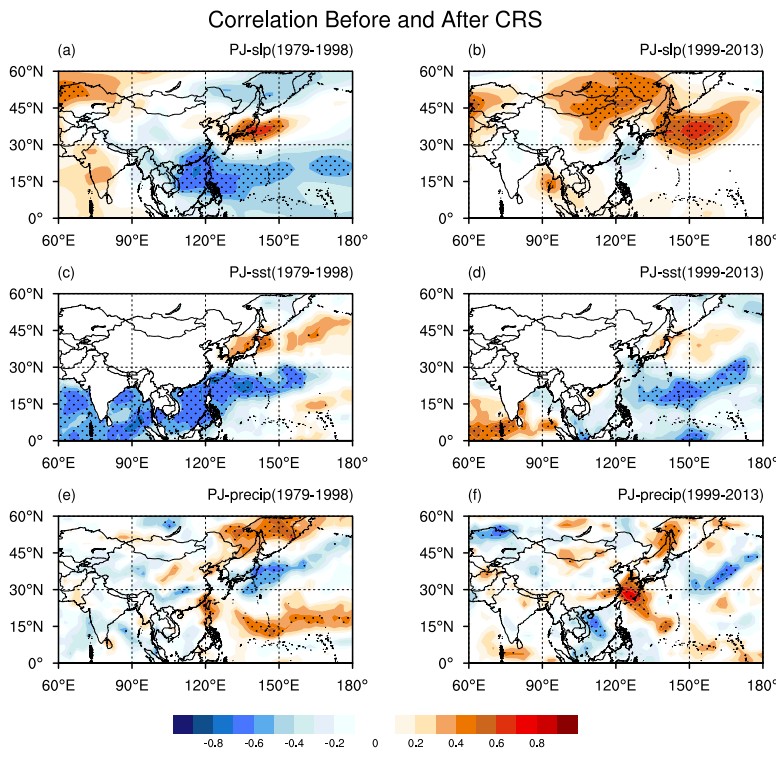

**Figure 6.** Correlation maps of the PJ index before (left panel) and after (right panel) the year 1999. The top, centre, and bottom panels are PJ vs. SLP, PJ vs. ERSST, and PJ vs. GPCP data, respectively. Correlation values at a 5% significance level are stippled.





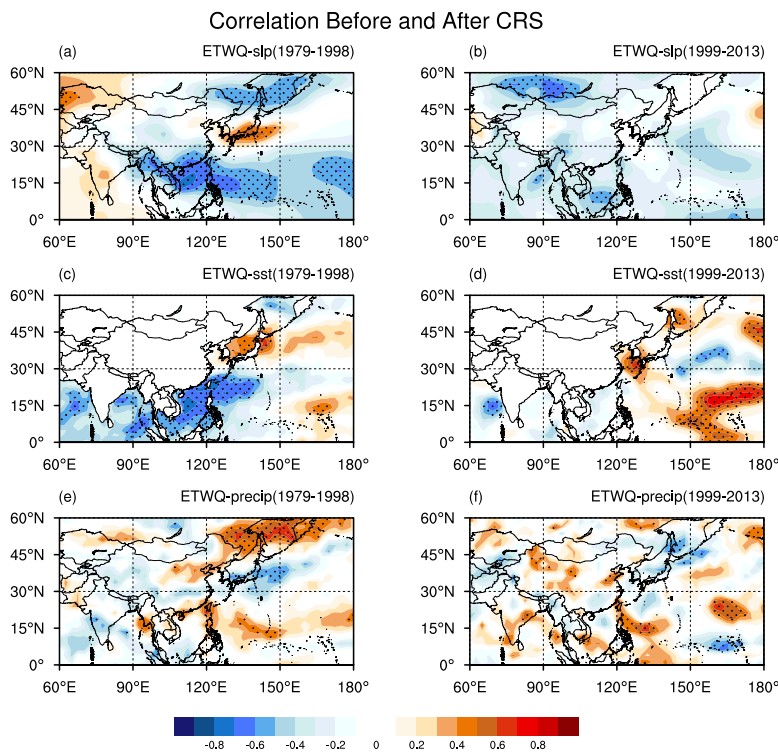

**Figure 7.** Correlation maps of the East Taiwan runoff (ETWQ) before (left panel) and after (right panel) the year 1999. The top, centre, and bottom panels are ETWQ vs. SLP, ETWQ vs. ERSST, and ETWQ vs. GPCP data, respectively. Correlation values at a 5% significance level are stippled.



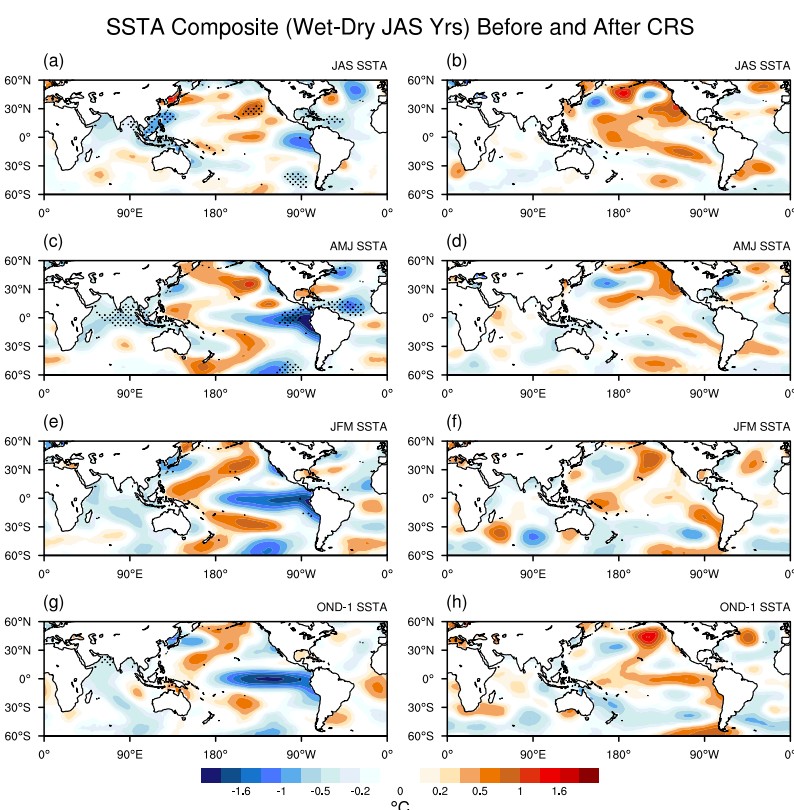

**Figure 8.** Evolution of SSTA composites from 0- to multi-month lead times based on wet-minus-dry years for the JAS ETWQ before (left panel) and after (right panel) the year 1999. Stippled areas indicate that the SSTA difference is at a 5% significance level according to Student's $t$-test.





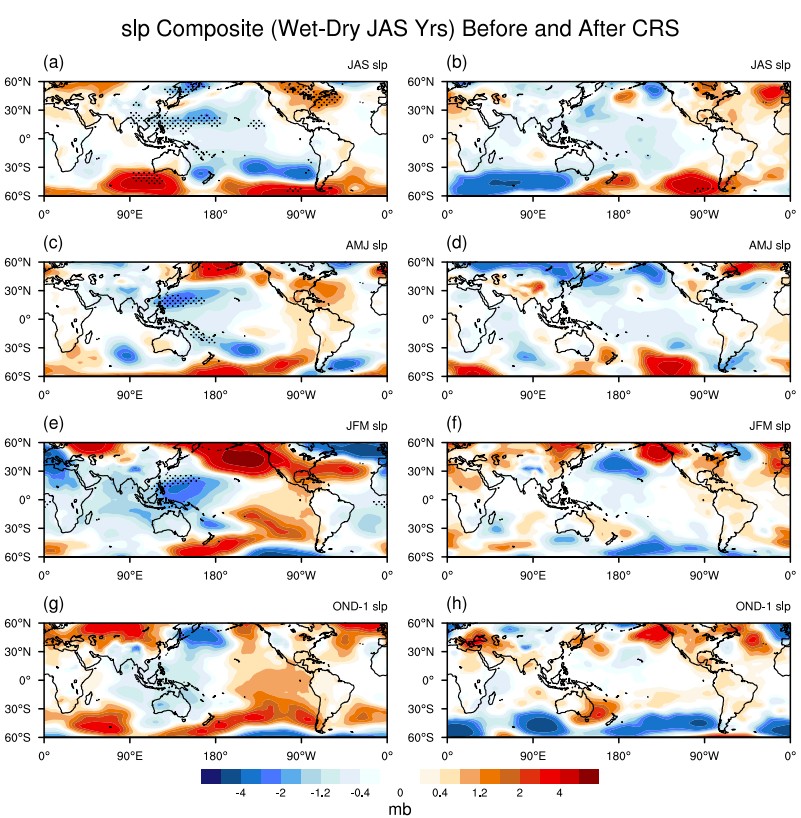

**Figure 9.** As in Figure 8, but for sea level pressure (slp) anomaly composites.



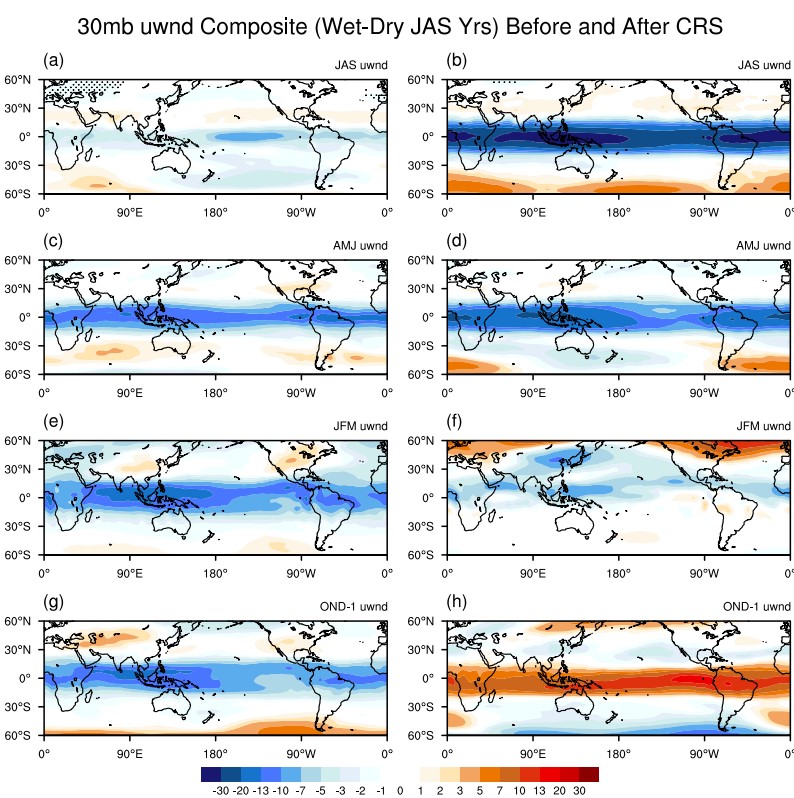

**Figure 10.** As in Figure 8, but for 30-mb zonal wind (uwnd) anomaly composites.