# Peer review of "Variations in the Correlation between Teleconnections and Taiwan's Streamflow"

_Hydrology and Earth System Sciences, 2017_

## Referee Comment (RC1) · Anonymous Referee #1 · 20 Mar 2017

Having previously reviewed an earlier draft of this manuscript, it was a pleasure to be able to review it again with the previous reviewer's comments addressed. While the revised article is not much different to the previous version, it is no longer part of a special issue on seasonal forecasting, and I feel it fits better as an individual contribution to HESS.

In general, this paper offers something novel and interesting to the scientific community, however I do - as before - still have some issues with the way the paper has been written. The language and writing style has not really been addressed since the previous version, and I think it would benefit from being much tighter and succinct in part.

[Figure]

For example, the beginning of the discussion is overly wordy and a bit conversational. I would suggest it be altered from:

"In this section, further discussion is provided to address the issue of CRS and how it can impact the convention of seasonal forecasting evidenced by some large-scale patterns. To start the discussion, we would like to argue that the two change points found for the response of Taiwan's streamflow to large-scale circulations are not a coincidence."

to something like this:

"In this study, we find that the two change points for the response of Taiwan's stream-flow to large-scale circulations are not a coincidence."

As the discussion moves forward, you could then bring in the issue of CRS and how it can impact seasonal forecasting evidenced by some large-scale patterns. This is just one example, and there are many parts of the manuscript that I believe could be improved along these lines.

On the topic of seasonal forecasting itself, this is first introduced at the very end of the introduction (as the authors note the following structure of the paper), but there is no explanation for its inclusion. It is discussed in section 4, but why not before then if this is truly a motivation of the paper? I feel that this is a bit of a hangover from the previous version that was then part of a special issue on seasonal forecasting, but it now feels out of place and perhaps a little bit forced.

In conclusion, I believe this paper to be worthy of publication, but would just like to see it tightened a little before it is published.

---

## Referee Comment (RC2) · Anonymous Referee #2 · 23 Mar 2017

The purpose of this paper is to explore possible usage of some well documented and routinely updated teleconnection indices to predict the high-flow season (July-September) 3-month streamflow totals in Taiwan. The WP (West-Pacific), PJ (Pacific-Japan) and the QBO (Quasi-Biennial Oscillation) are identified as of the highest correlation with Taiwan streamflow among 14 indices. The authors intended to elaborate a point that despite high correlations, the non-stationary behavior of PJ and the streamflow time series hampers the predictability of streamflow seasonal forecast. It seems to me that the conclusion is the teleconnection indices or any predictor with CRS are of no use to Taiwan JAS streamflow prediction. Unfortunately, the authors failed to propose any alternative solution to overcome the problem, so the paper appears not yet

ready for publication.

I suggest the authors to continue the research and take the following comments into account.

1. It is better not to use "relationship" in the title. Most discussion on "relationship" is more or less hand-waving with no evidence. For example, the discussion about possible relationship between QBO and Taiwan's streamflow on Page 10 Line 13-21 is based on possible influence of QBO on the total number of TCs (Chan 2003) and TC tracks (Ho et al. 2009) without presenting any evidence to support the relationship between Taiwan's streamflow variability and western North Pacific TC number or tracks. Similar weakness can be found in many places when the "relationship" is discussed.

2. Is QBO a better predictor than PJ? The correlation of PJ and Taiwan streamflow is highly influence by CRS but the correlation of QBO and the streamflow seems quite stable?

3. P11L19: Why is PDO selected most frequently as a predictor?

4. The last paragraph in the Summary and Conclusion is hard to understand. What is the "new predictor screening algorithm capable of accounting for CRSs"? Is the concept discussed in earlier paragraphs?

5. The new findings in this paper need to be sharpened and writing need to be more exact and concise.

---

## Author Response (AR1)

1 June 1, 2017

Professor Alberto Guadagnini
Executive Editor
Hydrology and Earth System Sciences

Dear Professor Guadagnini:

This manuscript has undergone three rounds of review.    My co-author and I thank all the referees for their comments and suggestions, and we believe the manuscript now is a polished one.    In the last round of review, we really appreciate Referee #1 who gave very positive evaluation and suggested the manuscript is worthy of publication in HESS. We are also very thankful to Referee #2 for the constructive comment regarding seeking a solution to streamflow forecasting.    However, it is very clear that the establishment of a forecasting tool is beyond the scope of this paper.    None of the previous studies have analyzed the correlations between Taiwan's streamflow and teleconnection indices; we believe such analysis is a prerequisite for the forecasting tool.    To avoid a lengthy manuscript, we skip the forecasting part in the main text, while the prototype linear regression model is shown in the Supplementary Material.    We are actually working on our next manuscript regarding the forecasting utility based on the findings from the current study.    In closing, we wish to draw your attention that we have considerably rewritten and rearranged many parts of the manuscript to make it more concise.

We hope the referees will be satisfied with the revised manuscript.

Sincerely yours,

Tsung-Yu Lee, Ph.D.
Dept. of Geography, National Taiwan Normal University

**Interactive Comment* on "An Investigation into the Relationship between Teleconnections and Taiwan's Streamflow" *by* Chia-Jeng Chen and Tsung-Yu Lee**

**Anonymous Referee #1**

Having previously reviewed an earlier draft of this manuscript, it was a pleasure to be able to review it again with the previous reviewer's comments addressed. While the revised article is not much different to the previous version, it is no longer part of a special issue on seasonal forecasting, and I feel it fits better as an individual contribution to HESS.

In general, this paper offers something novel and interesting to the scientific community, however I do - as before - still have some issues with the way the paper has been written. The language and writing style has not really been addressed since the previous version, and I think it would benefit from being much tighter and succinct in part.

Thank you for your re-review and constructive comments on the writing of the paper. While we have used the language editing service to fix some minor language issue for the previous version of the article, we realize that we have to alter the writing style to make the paper much tighter and less conversational.

In this revision, we have made our greatest effort to sharpen the writing to meet the publication standard. We wish you will find the revision satisfactory and support the final publication decision.

For example, the beginning of the discussion is overly wordy and a bit conversational. I would suggest it be altered from:

"In this section, further discussion is provided to address the issue of CRS and how it can impact the convention of seasonal forecasting evidenced by some large-scale patterns. To start the discussion, we would like to argue that the two change points found for the response of Taiwan's streamflow to large-scale circulations are not a coincidence."

to something like this:

"In this study, we find that the two change points for the response of Taiwan's streamflow to large-scale circulations are not a coincidence."

As the discussion moves forward, you could then bring in the issue of CRS and how it can impact seasonal forecasting evidenced by some large-scale patterns. This is just one example, and there are many parts of the manuscript that I believe could be improved along these lines.

We have revised the beginning of the discussion as suggested. As you mentioned, we also find that we can improve many parts of the paper along these lines. As another example, we have altered the beginning of the result section from

> *"In line with the aforementioned instruction, the correlation analysis is conducted for all the target gauges in Taiwan. Because the total number of combinations of the different gauges (upstream and downstream), climate indices, and lagged periods is in the thousands, the resulting correlation values are merely too many to be fully listed here. Therefore, the results are presented in a selective and illustrative fashion."*

to

> *"We selectively illustrate the correlation values from the many combinations of the different gauges, climate indices, and lagged periods."*

As a result, we have made a considerable reduction in word counts (>1500 words).

On the topic of seasonal forecasting itself, this is first introduced at the very end of the introduction (as the authors note the following structure of the paper), but there is no explanation for its inclusion. It is discussed in section 4, but why not before then if this is truly a motivation of the paper? I feel that this is a bit of a hangover from the previous version that was then part of a special issue on seasonal forecasting, but it now feels out of place and perhaps a little bit forced.

Totally agreed. This is truly a hangover from the previous version due to some earlier revisions in response to the special issue.

To accommodate this issue, we have removed the third objective and corresponding sentence from the end of the introduction (as well as in the abstract). We have moved Sections 2.4 (Linear regression prediction model) and 3.3 (Variations in prediction skill due to the shifts) and Figure 5 to the supplemental material as an experiment, and briefly mention the experiment at the end of Section 3.2 in case some readers might be interested in seeing how variations in prediction skill can be affected by the CRSs.

In conclusion, I believe this paper to be worthy of publication, but would just like to see it tightened a little before it is published.

Thank you very much again for your positive appraisal of our work.

***Interactive Comment*** on "An Investigation into the Relationship between Teleconnections and Taiwan's Streamflow" *by* Chia-Jeng Chen and Tsung-Yu Lee

**Anonymous Referee #2**

The purpose of this paper is to explore possible usage of some well documented and routinely updated teleconnection indices to predict the high-flow season (July-September) 3-month streamflow totals in Taiwan. The WP (West-Pacific), PJ (Pacific-Japan) and the QBO (Quasi-Biennial Oscillation) are identified as of the highest correlation with Taiwan streamflow among 14 indices. The authors intended to elaborate a point that despite high correlations, the non-stationary behavior of PJ and the streamflow time series hampers the predictability of streamflow seasonal forecast. It seems to me that the conclusion is the teleconnection indices or any predictor with CRS are of no use to Taiwan JAS streamflow prediction. Unfortunately, the authors failed to propose any alternative solution to overcome the problem, so the paper appears not yet ready for publication.

I suggest the authors to continue the research and take the following comments into account.

Thank you for your review and comments.

We realize that the reviewer's principal objection to our paper is the lack of an alternative solution or in fact, a new method to overcome the problem of streamflow prediction using predictor(s) with CRSs. To begin with our response, we would like to argue that the new method (currently under preparation) is better to be a stand-alone contribution to a follow-up article. The reasons for opting out of the new prediction method are mainly twofold, in terms of:

1) the background of this article and the shaped study scope:
   The earlier version of this article was aimed at the discussion about the correlation between teleconnections and Taiwan's streamflow. The article was incidentally transferred to the special issue of seasonal forecasting, so some forecasting elements, by necessity, were included in response to previous review comments. Now this article is resubmitted as an ordinary contribution to HESS. We would like to restate the original scope of this article—a teleconnection paper—rather than a prediction paper. To also sharpen our findings and tighten the writing, we have trimmed those hangover prediction elements in the paper for this round of revision. In addition, the new method to overcome the prediction problem dealing with CRSs is really our ensuing work. If we introduced the new idea and included more pertinent assessments in this paper, we are afraid that the paper will be lengthy and defocused.

2) the new method in substance:
   In the conclusion section, we have already indicated that the new prediction method is under way, which is also noted by the reviewer in your specific comment 4.  To further explain why the inclusion of the new method may steel away from the current research theme, we believe that we should disclose some ideas regarding the new method itself.

In the present paper, we have shown that varied predictive relationships can be seen during different phases of the PDO (i.e., CRS over the Pacific).  In the context of predictor screening, our finding suggests that a new set of predictors should be identified when the PDO changes to a new phase that can potentially last for several decades.  Nevertheless, it is impossible to do so without the effective forecasting of the phase change in the PDO with extended lead times.  This is one of the toughest, unsolved problems in the scientific community; even the state-of-the-art GCMs have hard time resolving the correct variations in the climate indices in hindcasting experiments.  Therefore, using those identified climate indices as predictors seems unrealistic.  A remedy for this problem, as well as a cornerstone of our new prediction model, relies on the predictor screening algorithm, established by our first author (Chen and Georgakakos, 2014), capable of identifying new predictors from oceanic and/or atmospheric fields and accounting for the effect of CRSs on predictor screening. The present paper and associated findings do serve as the backbone and motivation of the development of the new model that is in our ongoing manuscript.

In this revision, we have carefully taken your following comments in to account and mainly worked on the writing part to make the paper more exact and concise. We wish you find our response and revision satisfactory, and look forward to your re-evaluation.

[References]

Chen, C-J. and Georgakakos, A. P., 2014: Hydro-climatic forecasting using sea surface temperatures—methodology and application for the Southeast U.S. *Clim. Dyn.*, **42**, 2955–2982.

1) It is better not to use "relationship" in the title.  Most discussion on "relationship" is more or less hand-waving with no evidence.  For example, the discussion about possible relationship between QBO and Taiwan's streamflow on Page 10 Line 13-21 is based on possible influence of QBO on the total number of TCs (Chan 2003) and TC tracks (Ho et al. 2009) without presenting any evidence to support the relationship between Taiwan's streamflow variability and western North Pacific TC number or tracks.  Similar weakness can be found in many places when the "relationship" is discussed.

The title has been changed to "Variations in the Correlation between Teleconnections and Taiwan's Streamflow." We also avoid the use of "relationship" in the text when causal mechanisms are not evident.

Nevertheless, we respectfully disagree that your specific comment on our discussion is hand-waving with no evidence. Studies (e.g., Chen et al., 2010; Kao et al., 2011; Tu and Chou, 2013; Liang et al., 2017) have clearly indicate that typhoon rains account for ~50% of the total annual rainfall and ~75% of the annual discharge in Taiwan, so the variability in TC number and tracks that influences the variability in Taiwan's rainfall certainly shows influence on the variability in Taiwan's streamflow.

[References]

Chen, J. M., Li, T., and Shih, C. F., 2010: Tropical cyclone- and monsoon-induced rainfall variability in Taiwan. *J. Clim.*, **23**, 4107–4120.

Kao, S. J., Huang, J. C., Lee, T. Y., Walling, D. E., 2011: The changing rainfall–runoff dynamics and sediment response of small mountainous rivers in Taiwan under a warming climate. A chapter in *Sediment Problems and Sediment Management in Asian River Basins*, edited by Walling, D. E., pp. 114–129.

Tu, J. Y. and Chou, C., 2013: Changes in precipitation frequency and intensity in the vicinity of Taiwan: typhoon versus non-typhoon events. *Environ. Res. Lett.*, **8**, 014023.

Liang, A., Oey, L., Huang, S., and Chou, S., 2017: Long-term trends of typhoon-induced rainfall over Taiwan: In situ evidence of poleward shift of typhoons in western North Pacific in recent decades. *J. Geophys. Res.*, **122(5)**, 2750–2765.

2) Is QBO a better predictor than PJ? The correlation of PJ and Taiwan streamflow is highly influence by CRS but the correlation of QBO and the streamflow seems quite stable?

Unfortunately, this is not true. The QBO also experiences the phase reversal before and after the CRS. We have already addressed that in the discussion section (Section 4, Page 14) and shown the phase change in 30mb u-wind composites in Figure 10.

3) P11L19: Why is PDO selected most frequently as a predictor?

This is because we only show the "selective" hindcasting results using linear regression in Section 3.3. If you look up Table 3 for the correlation values with the PDO index, you can find that many of them are significant at $p = 0.05$. Those

catchments shown in Figure 5 coincidentally have JAS flow well correlated with the PDO index. Based on the AIC and the VIF model selection criteria, it is reasonable to have the PDO be selected most frequently in this case.

Please note that in response to your comment 5 below (i.e., to sharpen the new findings and tighten the writing), we have moved Sections 2.4 (Linear regression prediction model) and 3.3 (Variations in prediction skill due to the shifts) and Figure 5 to the supplemental material as an experimental case study, and briefly mention the experiment at the end of Section 3.2 in case some readers might still be interested in seeing how prediction skill can vary because of the CRSs.

4) The last paragraph in the Summary and Conclusion is hard to understand. What is the "new predictor screening algorithm capable of accounting for CRSs"? Is the concept discussed in earlier paragraphs?

We have already addressed the related discussion about how CRSs adversely affect predictor screening in the discussion section (Section 4, Pages 13–14); however, in the present paper, we do not disclose how a new predictor screening algorithm should be implemented to be able to account for CRSs.

Again, the new algorithm belongs to one of our working items. To avoid any further misconception and tighten the writing, we have modified the last paragraph of the paper as:

> *"Our current endeavour includes applying a similar analysis framework to Taiwan's streamflow in other seasons, such as the spring rains (February to April) and Mei-Yu (May and June) seasons. Associated findings will be incorporated into a seasonal streamflow forecasting model to improve regional water resource planning and management."*

5) The new findings in this paper need to be sharpened and writing need to be more exact and concise.

Agreed. We have re-organized and re-written many sentences and paragraphs of the paper in a much tighter style. For example, according to the other reviewer's suggestion, we have altered the beginning of the discussion from

> *"In this section, further discussion is provided to address the issue of CRS and how it can impact the convention of seasonal forecasting evidenced by some large-scale patterns. To start the discussion, we would like to argue that the two change points found for the response of Taiwan's streamflow to large-scale circulations are not a coincidence."*

to

> *"In this study, we find that the two change points for the response of Taiwan's streamflow to large-scale circulations are not a coincidence."*

As another example, we have altered the beginning of the result section from

> *"In line with the aforementioned instruction, the correlation analysis is conducted for all the target gauges in Taiwan.  Because the total number of combinations of the different gauges (upstream and downstream), climate indices, and lagged periods is in the thousands, the resulting correlation values are merely too many to be fully listed here. Therefore, the results are presented in a selective and illustrative fashion."*

to

> *"We selectively illustrate the correlation values from the many combinations of the different gauges, climate indices, and lagged periods."*

As a result, we have made a considerable reduction in word counts (>1500 words).

[revised manuscript text omitted]